# Effectiveness of Live Attenuated Varicella-Zoster Vaccine in Adults Older than 50 Years in Japan: A Retrospective Cohort Study

**DOI:** 10.3390/vaccines11020259

**Published:** 2023-01-25

**Authors:** Kazuhiro Matsumoto, Satoko Ohfuji, Kana Inohara, Masateru Akechi, Hiroko Kumashiro, Motoki Ishibashi, Shin Irie, Yoshio Hirota

**Affiliations:** 1Department of Public Health, Osaka Metropolitan University Graduate School of Medicine, Osaka 545-8585, Japan; 2The Research Foundation for Microbial Diseases of Osaka University, Osaka 565-0871, Japan; 3SOUSEIKAI Medical Group, Fukuoka 813-0017, Japan

**Keywords:** herpes zoster, vaccine effectiveness, cohort study

## Abstract

*Background*: In Japan, freeze-dried live attenuated varicella-zoster vaccine BIKEN is available for adults aged ≥50 years to prevent herpes zoster (HZ). A prospective cohort study of 1200 healthy adults and 300 patients with underlying illness confirmed vaccine safety between 2016 and 2017. However, evidence of vaccine effectiveness (VE) is limited. *Methods*: VE against HZ and postherpetic neuralgia (PHN) was evaluated in the vaccinated cohort of the previous safety study in a follow-up study between 2021 and 2022 and compared with unvaccinated family members. Self-administered questionnaires determined retrospective experiences of HZ and PHN diagnosis. Logistic regression estimated the VE by calculating the outcome odds ratio (OR) in vaccinated vs. unvaccinated groups: VE = (1 − OR) × 100(%). *Results*: Overall, 1098 vaccinated and 518 unvaccinated subjects were analysed. Between 2016 and 2022, 26 vaccinated (2.4%) and 22 unvaccinated (4.2%) subjects reported HZ diagnosis, and 3 vaccinated (0.3%) and 2 unvaccinated (0.4%) subjects reported PHN. Adjusted VE against a clinical diagnosis was 41% for HZ [−6% to 67%], with marginal significance, and 16% [−408% to 86%] for PHN. Stratification by age, sex, or comorbidities had an adjusted VE against HZ of ~40%, which was similar between strata. *Conclusion*: Freeze-dried live attenuated varicella-zoster vaccine reduces the risk of HZ regardless of age, sex, or comorbidities.

## 1. Introduction

Primary varicella-zoster virus (VZV) infection causes varicella and results in latent infection. Herpes zoster (HZ), characterized by unilateral radicular pain and a vesicular rash, results from the reactivation of latent VZV in the sensory ganglia. It can expand to involve several dermatomes, especially in immunocompromised subjects. The frequency and severity of HZ increase with age, which correlates closely with a continuous decline in cell-mediated immunity to VZV [1]. Postherpetic neuralgia (PHN) is pain after an acute episode of HZ continuing beyond rash healing [2]. Symptoms can persist for months or even years, disturb sleep, or become exacerbated by contact of skin with garments. The condition can compromise the patient’s quality of life [3,4] and ability to function to a degree comparable to that in diseases such as congestive heart failure, myocardial infarction, diabetes mellitus type 2, and major depression [5]. Antiviral therapy reduces the severity and duration of herpes zoster but does not prevent the development of PHN [6,7]. PHN is often refractory to treatment [2]. PHN is the most common complication of HZ; an estimated 12.5% of patients with HZ aged ≥50 years have PHN 3 months after HZ onset, and the proportion increases sharply with age [8]. According to Japanese studies, the incidence of HZ among older individuals has been increasing to 10.2/1000 person-years [9] or 10.9/1000 person-years [10]. Another Japanese study indicated that the incidence of HZ among those aged ≥65 years was 12.3/1000 person-years [11]. In addition, patients with diabetes mellitus, autoimmune diseases, renal failure, and malignancies have a higher risk of HZ than those with other diseases [12,13]. In addition, the proportion developing PHN among HZ patients ranged from 9% [9] to 19% [10], and its risk was increased in males, age ≥65 years, and immunosuppressive therapy [9]. Thus, HZ and PHN have become major concerns in Japan and it is important to protect these high-risk populations from the threat of HZ and PHN. ZOSTAVAX^®^ (Merk & Co., Inc., Rahway, NJ, USA) is a live attenuated virus vaccine for the Oka strain of HZ (19,400 PFU or more, based on the package insert) and has been approved in more than 60 countries or counties for prophylactic use in older individuals. ZOSTAVAX^®^ was reported to increase VZV antibody titers and cell-mediated immunity, which have an important role in controlling the onset of HZ and PHN [14,15,16]. The clinical efficacy for reducing the incidence of HZ was reported to be 51.3% and 66.5% for reducing the incidence of PHN [17]. However, ZOSTAVAX^®^ has not been approved in Japan. Instead of ZOSTAVAX^®^, a freeze-dried live attenuated varicella vaccine by Research Foundation for Microbial Diseases of Osaka University (BIKEN) for the Oka strain (1000 PFU or more, based on the package insert), which was originally used in 1986 to prevent varicella in children, was additionally approved for use to reduce the risk of HZ in individuals aged ≥50 years in 2016. Because this varicella-zoster vaccine generally contains live attenuated Oka virus of 23,000–95,000 PFU [18], it is used to prevent not only varicella in children, but also HZ in adults in Japan. However, the data of its effectiveness in preventing HZ and PHN in the real-world setting are limited. Between 2016 and 2017, we conducted a prospective cohort study of 1200 healthy adults and 300 patients with high-risk conditions for HZ (JapicCTI-163415) and reported the safety of the varicella-zoster vaccine in healthy adults and high-risk groups [19]. Here, we followed up the participants of the previous safety study regarding the development of HZ and PHN as the vaccinated group and compared the incidence of their spouses as the unvaccinated group to evaluate the effectiveness of the varicella-zoster vaccine against HZ and PHN.

## 2. Materials and Methods

### 2.1. Study Design, Setting, and Participants

Using the database of the previous study [19], a retrospective cohort study was conducted to evaluate the effectiveness of the varicella-zoster vaccine against HZ and PHN between August 2021 and March 2022. The details of the previous study have been described elsewhere [19]. We asked previous study participants to join the present study of vaccine effectiveness as the vaccinated group. Only participants who were able to contact with us took part in the present study. The previous study participants included not only healthy adults, but also patients with high-risk conditions for HZ, such as malignancy, diabetes mellitus, autoimmune diseases, and chronic renal diseases [19]. Patients with malignancy included those with malignant solid tumor, such as colon cancer, lung cancer, gastric cancer, liver cancer, breast cancer, prostate cancer, cervical cancer, or with malignant lymphocytic leukemia, who were in the remission stage at the time of enrolment. Among them, the following patients were excluded: those who received immunosuppressive chemotherapy or radiation therapy within the preceding 6 months (or were planned to receive it within 28 days after vaccination); for patients with acute lymphocytic leukemia, those who had reached the remission stage within the preceding 3 months, those whose number of lymphocytic leukemia was less than 500/mm^3^, those with a negative result on the delayed skin hypersensitivity test, and those who received chemotherapy for remission maintenance using medications other than 6-mercaptopurine within the preceding 1 week (or were planned for it within 28 days after vaccination); and, for patients with malignant solid tumor, those whose tumor development could not be controlled by surgery or chemotherapy and those whose tumor development was under control but who received immunosuppressive chemotherapy or radiation therapy within the preceding 6 months (or were planned for it within 28 days after vaccination). The inclusion criteria for diabetes mellitus patients were those diagnosed with diabetes mellitus; those without diabetic neuropathy; those whose diabetes was not caused by the side effects of immunosuppressants; and those who did not receive cortical hormones, immunosuppressants, or antiplatelet therapy, including aspirin. Regarding autoimmune diseases, patients with rheumatoid arthritis, systemic lupus erythematosus, collagen diseases, and ulcerative colitis were candidates for enrolment. Among them, patients who received cortical hormones, immunosuppressants, biologic agents, or JAK inhibitors within the preceding 6 months were excluded. Patients with chronic renal diseases were regarded as those with findings compatible with renal disease diagnosed by urinalysis, imaging, laboratory, or pathological examination. For example, patients whose albuminuria (≥30 mg/gCr) or proteinuria (≥0.15 g/gCr) had continued for ≥3 months or those with eGFR levels of 46–59 mL/min/1.73 m^2^ were included. Patients receiving cortical hormones or immunosuppressants were excluded. Healthy adults >50 years were also enrolled. Those with mild underlying illness, such as hypertension and dyslipidemia, if well controlled, were allowed to participate.

To compare the incidence of HZ and PHN in these vaccinated subjects with unvaccinated subjects, we invited their family members, whose lifestyles were similar, to join the present study of vaccine effectiveness. Because some subjects might have a history of varicella zoster vaccination, we confirmed the accurate varicella-zoster vaccination status of the candidate family members based on information from a self-administered questionnaire. All participants were Japanese adults ≥50 years, irrespective of having an underlying illness or not, and provided written informed consent. Exclusion criteria included the receipt of the recombinant zoster vaccine.

This study was performed in accordance with the Declaration of Helsinki and the study protocol was approved by the ethics committees at Global Clinical Research Center, SOUSEIKAI, Hakata clinic (1668CP-2).

### 2.2. Information Collection

We collected demographic information, including age, sex, history of any diseases, comorbidities, name of medication, history of varicella-zoster vaccination, symptoms of HZ (the location of rash, severity of pain, duration of symptoms, and current condition), and a diagnostic history of HZ using a self-administered questionnaire. The severity of pain was determined by 6 ranked categories (no pain, very mild, discomforting, tolerable, intense, and very intense) and we recategorized these into 3 levels in the analysis: mild (no pain or very mild), moderate (discomforting or tolerable), and severe (intense or very intense). The duration of pain was also determined by 6 ranked categories (less than a week, less than 2 weeks, less than 3 weeks, less than a month, less than 2 months, and more than 3 months) and we recategorized these into 3 levels in the analysis: short (less than 2 weeks), moderate (less than a month), and long (more than a month). If there was a diagnostic history of HZ, we asked the medical institution where HZ was diagnosed to fill in a questionnaire about the details of the disease, including the date of the diagnosis, application of a rapid examination kit, symptoms, sequela (PHN, meningitis, corneal ulcer, Ramsay Hunt syndrome, paralysis, bladder bowel disorder, and others), and treatment (antiviral therapy and analgesia).

### 2.3. Vaccination

In the previous study (between 2016 and 2017), all vaccinated participants received one subcutaneous injection of 0.5 mL live attenuated varicella (Lot Nos. VZ184, 189, 200) manufactured by BIKEN the Research Foundation for Microbial Diseases of Osaka University. Each vaccine was supplied as a single-dose viral containing live attenuated Oka varicella-zoster virus (29,000–58,000 PFU). No adjuvant was included in the vaccine.

### 2.4. Outcomes

The primary objective was to evaluate the overall vaccine effectiveness in reducing the risk of HZ and PHN. A clinical diagnosis of HZ was determined by a self-administered questionnaire. PHN was defined as any long-lasting pain (more than 3 months or remaining when answering the questionnaire) with the clinical diagnosis of HZ.

### 2.5. Statistical Analyses

The characteristics of the vaccinated and unvaccinated groups were examined by χ2 test or Wilcoxon’s rank sum test. To estimate VE against HZ, Cox proportional hazard model was firstly employed. The follow-up period was between April 2017 and the onset of HZ or the enrollment of the present study, whichever came first. In this model, we tested the proportional hazards assumption. If *p*-value suggested that proportionality was not satisfied, we created a Kaplan–Meier curve to examine the applicability of Cox proportional hazard model. As a result, Cox proportional hazard model was not applicable. Therefore, we decided to use logistic regression model. In the logistic regression model, we calculated the outcome odds ratio (OR) in vaccinated vs. unvaccinated and estimated VE by the following formula: VE = (1 − OR) × 100(%). In multivariate models, we controlled for sex, age (<70/≥70 years), and comorbidities (no/yes). Comorbidities include various diseases, such as hypertension, dyslipidemia, diabetes mellitus, chronic renal diseases, autoimmune diseases, respiratory diseases, malignant solid tumor, and other diseases. We also conducted analysis separately controlled for each comorbidity instead of all comorbidities. In addition, we performed additional stratified analyses to consider the effect of confounding variables, including sex, age (70</≥70 years), and comorbidities. The subjects were divided into two groups according to the variables and the estimated VE of each group. The homogeneity of VE across stratified categories was tested as the *p*-value of the interaction term between vaccination and each stratified variable. Statistical analyses were conducted using SAS version 9.4. A *p*-value < 0.05 or a positive lower bound of the confidence interval (CI) for VE was considered to indicate statistical significance, and a *p*-value < 0.10 was considered as marginal significance.

## 3. Results

### 3.1. Characteristics of the Subjects

A total of 1616 participants were enrolled, of whom 1096 were previous study participants and 520 were family members (461 spouses, 14 brothers or sisters, and 4 children, and 41 others). Two of the family members were found to be vaccinated based on the self-administered questionnaire. Eventually, 1098 vaccinated and 518 unvaccinated subjects were analyzed. The characteristics of both groups are shown in Table 1. The median age and the proportion of those aged ≥70 years were not significantly different between the vaccinated and unvaccinated groups. The proportion of males was higher in the vaccinated group than in the unvaccinated group. The proportions of comorbidities and medication were higher in the unvaccinated group than in the vaccinated group. The median follow-up period of all subjects was 51 months.

### 3.2. Clinical Outcomes of the Subjects

Between 2016 and 2022, a total of 48 subjects (26 vaccinated and 22 unvaccinated) reported a clinical diagnosis of HZ and 5 subjects (3 vaccinated and 2 unvaccinated) suffered from PHN (Table 2). Among 48 subjects with self-reported HZ, the medical records of 26 patients were available (16 vaccinated and 10 unvaccinated). All patients whose medical records were available underwent a rapid diagnostic test examination. The consistency between the self-administered questionnaire and medical records was 81% in the vaccinated group and 90% in the unvaccinated group (Appendix A). A total of six subjects experienced sequela (five subjects reported PHN and the other reported nonspecific discomfort). Twelve subjects in the vaccinated group and eight subjects in the unvaccinated group received antiviral therapy.

### 3.3. Overall VE against HZ and PHN

According to Cox proportional hazard model, the adjusted VE against HZ was 38% (−10 to 66%) but the proportionality was not satisfied (*p* = 0.03). The overall VE against HZ and PHN by using a logistic regression model is shown in Table 3. The crude vaccine effectiveness against HZ was 45% (95% CI; 3 to 69%) with statistical significance and the adjusted VE was 41% (−6 to 67%) with marginal significance (*p* = 0.08). Even after separately adjusting for each comorbidity instead of all comorbidities, the VEs were not so changed. Against PHN, the crude and adjusted VE were 29% and 16%, respectively, which were statistically insignificant. More than half of the subjects in both groups reported the severity of pain was moderate and there was no statistical difference between the vaccinated group and the unvaccinated group. Furthermore, 61% of the vaccinated group and 73% of the unvaccinated group reported the duration of pain was short. Only 8% of the vaccinated group and 4% of the unvaccinated group experienced pain longer than a month (Appendix A).

### 3.4. VE against HZ in the Stratified Analyses

The VE in each group is shown in Table 4. The adjusted VE was approximately 40% and did not differ between subjects with or without comorbidities, between sexes, or between age categories.

## 4. Discussion

In the present study, the crude overall effectiveness of freeze-dried live attenuated varicella vaccine against HZ was 45% with statistical significance. However, the adjusted VE was 41% and could not reach statical significance. The VE was similar regardless of comorbidities, sex, and age groups and also statistically insignificant. The reason might be small sample size.

A double-blind, placebo-controlled trial in which 38,546 adults aged ≥60 years were randomized to receive a single subcutaneous dose of Zostavax^®^ or placebo demonstrated that the overall efficacy against HZ was 51.3% [14]. According to another randomized placebo-controlled trial of 22,439 persons aged 50–59 years, Zostavax efficacy for the incidence of HZ was 69.8% during a mean follow-up of 1.3 years [15]. Observational studies indicated that the VE of Zostavax against HZ was between 48% and 55% in the general population [20,21,22]. The overall VE of the present study was slightly lower compared with these studies. One reason may be that the number of subjects in the present study was smaller than in other studies and, thus, the VE in the present study may have fluctuated.

The VE of Zostavax against PHN was reported to be between 59% and 66.5% [14,21]. A case–control study demonstrated that the vaccine reduced the risk of PHN by 41% among those with HZ [23]. This suggests that HZ vaccination provides incremental benefits beyond simply reducing HZ incidence. However, the VE against PHN in the present study was lower than in these studies. Although the number of PHN patients was too small to calculate meaningful VE in the present study, most of the benefit of the vaccine might be attributed to a reduction in HZ incidence rather than the PHN incidence.

In a large population-based cohort, the VE against HZ did not differ between individuals with diabetes mellitus and the general population (50% and 48%, respectively) [24]. According to a cohort study of adults ≥60 years who received chemotherapy, the adjusted VE was 42% [25]. A randomized controlled study demonstrated that the estimated VE against HZ in patients with solid tumor malignancies was 63.6% [26]. In the present study, the presence of any comorbidities did not influence the VE in the stratified analysis. Taken together, the impact of an underlying illness on the effectiveness of freeze-dried live attenuated varicella vaccine might be small.

A cohort study in the United Kingdom provided evidence that the VE was similar across sex and age groups [27]. This result was consistent with the present study. However, according to a randomized controlled study, immune responses after vaccination decreased with age and were significantly lower in subjects ≥70 years compared with subjects 60–69 years [18] and another cohort study reported that the VE gradually waned after vaccination [21]. Therefore, further studies are needed to examine the long-term effectiveness of varicella vaccines for the elderly.

This study had several limitations. First, we included only participants of the previous study who were able to respond to us. This sampling might have created selection bias because patients who had actually suffered HZ/PHN might be more interested in taking part in the follow-up study. On the other hand, the number of those who had died early and were not included in the present study was only seven. Therefore, the selection bias derived from early death might be small. Second, we have used relatives of the previous participants as the control group. Most relatives were the participants’ spouses. Therefore, the control group might not be matched to the case group by sex. However, their age and lifestyles were similar. Third, we depended on outcome information from self-administered questionnaires, which might have caused misclassifications. However, we used medical records to validate the diagnosis of HZ. Concordance between self-administered questionnaires and medical records was high (81% in the vaccinated group and 90% in the unvaccinated group). Fourth, the accuracy of diagnoses was a concern because each subject was diagnosed in a different medical institution. However, all subjects with available medical records were examined by a rapid examination kit. Moreover, concordance between the rapid identification kit and RT-PCR was 99.3% [28]. It is, therefore, considered that the inconsistency of diagnosis between medical institutions would be negligible. Fifth, to estimate VE, we mainly used a logistic model instead of Cox proportional model because the proportional hazard assumption was violated. However, the adjusted VE against HZ was similar in both models. Sixth, the present study was an extension of a previous study. The sample size might be small to evaluate vaccine effectiveness. In particular, we could not estimate VE against PHN precisely due to few outcomes. Seventh, vaccinated people who might be interested in their own health tended to recall their comorbidities and medication in detail. Therefore, there might be recall biases. Finally, observational studies can be subject to biases because patients who seek the vaccine may differ in their underlying risk of disease or in their ability and desire to access care for the condition [20].

## 5. Conclusions

In the multivariate model, the adjusted VE was 41% (−6 to 67%) with marginal significance (*p* = 0.08). In the stratified analysis, the adjusted VE was approximately 40% and did not differ between subjects with or without comorbidities, between sexes, or between age categories. These results suggest that the varicella-zoster vaccine BIKEN was likely to reduce the risk of HZ by about 40%, regardless of age, sex, or comorbidities.

## Figures and Tables

**Table 1 vaccines-11-00259-t001:** Characteristics of the study subjects.

Variables	Vaccinated Group (*n* = 1098)	Unvaccinated Group (*n* = 518)	*p*-Value ^a^
Age (years)			
Median [range]	67 (53–89)	67 (50–89)	0.96
Age (years)			
Older than 70 years	436 (40)	197 (38)	0.52
Male	563 (51)	220 (42)	<0.01
Comorbidities ^b^	472 (43)	266 (51)	<0.01
Hypertension	299 (27)	164 (32)	0.06
Dyslipidemia	112 (10)	49 (9)	0.64
Diabetes mellitus	127 (12)	69 (13)	0.31
Chronic renal diseases	4 (0)	14 (3)	<0.01
Autoimmune diseases ^c^	5 (0)	12 (2)	<0.01
Respiratory diseases ^d^	15 (1)	15 (3)	0.04
Malignant solid tumor	22 (2)	19 (4)	0.04
Other diseases	58 (5)	39 (8)	0.08
Medication	477 (43)	268 (52)	<0.01

Data are expressed as *n* (%) unless otherwise indicated. ^a^ Chi-square test, Fisher’s exact method, or Wilcoxon rank sum test were used as appropriate. ^b^ Includes hypertension/dyslipidemia/diabetes mellitus/chronic renal diseases/autoimmune diseases/respiratory diseases/malignant solid tumor/other diseases. ^c^ Includes rheumatoid arthritis/collagen disease/Crohn’s disease/ulcerative colitis/systemic lupus erythematosus. ^d^ Includes chronic bronchitis/emphysema/asthma.

**Table 2 vaccines-11-00259-t002:** Clinical diagnosis of the study subjects.

Variables	Vaccinated Group (*n* = 1098)	Unvaccinated Group (*n* = 518)	*p*-Value ^a^
HZ	26 (2.4)	22 (4.2)	0.04
PHN	3 (0.3)	2 (0.4)	0.66

Data are expressed as *n* (%) unless otherwise indicated; ^a^ Chi-square test or Fisher’s exact method were used as appropriate. HZ: Herpes zoster; PHN: Postherpetic neuralgia.

**Table 3 vaccines-11-00259-t003:** Overall vaccine effectiveness against HZ and PHN.

Outcome	No. of Outcome in Vaccinated Group (*n* = 1098)	No. of Outcome in Unvaccinated Group (*n* = 518)	Crude VE (95% CI)	Adjusted VE (95% CI) ^a^
HZ	26 (2.4)	22 (4.2)	45% (3–69%)	41% (−6 to 67%)
PHN	3 (0.3)	2 (0.4)	29% (−324 to 88%)	16% (−408 to 86%)

VE: vaccine effectiveness; CI: confidence interval; HZ: herpes zoster; PHN: postherpetic neuralgia. Data are expressed as *n* (%) unless otherwise indicated; ^a^ Adjusted variables: sex, age (<70/≥70 years), and comorbidities.

**Table 4 vaccines-11-00259-t004:** Vaccine effectiveness against HZ in the stratified analyses.

Stratified Category	Number of Outcome/Vaccinated Group (%)	Number of Outcome/Unvaccinated Group (%)	Crude VE (95% CI)	Adjusted VE (95% CI) ^a^	*p* for Interaction
Comorbidities (−)	14/626 (2)	11/252 (4)	50% (−12 to 78%)	40% (−36 to 73%)	0.82
Comorbidities (+)	12/472 (3)	11/266 (4)	40% (−39 to 74%)	39% (−42 to 73%)	
Male	9/563 (2)	6/220 (3)	42% (−65 to 80%)	34% (−91 to 77%)	0.98
Female	17/535 (3)	16/298 (5)	42% (−16 to 71%)	40% (−22 to 70%)	
<70 years	17/662 (3)	15/321 (5)	46% (−9 to 74%)	43% (−56 to 79%)	0.53
≥70 years	9/436 (2)	7/197 (4)	38% (−28 to 70%)	40% (−67 to 78%)	

^a^ Adjusted for the following potential confounders other than stratified factor: comorbidities, sex, and age (<70/≥70 years).

## Data Availability

The data presented in this study are available on request from the corresponding author. The data are not publicly available due to ethical reasons.

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
