# Peer review of "Effectiveness of Live Attenuated Varicella-Zoster Vaccine in Adults Older than 50 Years in Japan: A Retrospective Cohort Study"

_vaccines, 2023, doi:10.3390/vaccines11020259_

Round 1

Reviewer 1 Report

This is a retrospective observational study with a case-control design, regarding the effectiveness of a live Varicella-Zoster vaccine (freeze-dried live attenuated varicella-zoster vaccine BIKEN) in preventing herpes zoster (HZ) and post-herpetic neuralgia (PHN). It is noted that this vaccine is manufactured and used in Japan.

The study is an extension of a previous study regarding the safety of the same vaccine which has been published in 2019.

The manuscript is well-written, and the various sections are balanced in length.

INTRODUCTION

Introduction is concise and informative with appropriate references.

METHODS

In the methods section there are some issues to be addressed:

·        The original safety study had included 1500 participants. Of these, the authors have included in the present study 1096. It is not clear whether there was a selection process, or it was a convenience sample, e.g., the authors included all patients which were able to contact. The authors need to clarify this as the method  (if there was one) of selection might have created a biased sample. E.g., patients who had actually suffered HZ/PHN might be more interested in taking part on the follow-up study. Patients who have died early and were not included in the present study might have a smaller probability of presenting HZ or PHN since the follow-up time was shorter.

·        The selection of the control group is controversial, as the authors have used relatives of the participants. If possible, the control group should be matched to the case group, e.g., by age.

RESULTS

In the results section the following issues should be addressed:

·        A significant issue inherent with the case-control design, is whether the vaccinated and unvaccinated cohorts are similar in terms of known risk factors for HZ/PHN.

·        The authors have provided adjusted estimates of vaccine efficacy for age, sex, and comorbidities, but they have pooled all comorbidities together. However, not all underlying diseases confer the same risk of HS/PHN and patients with a single low risk comorbidity are considered similar to patients with multiple high-risk comorbidities (e.g., diabetes, CKD and malignancy). It is noted that, in the original study, the authors have included participants with high-risk conditions for HZ, such as malignancy, diabetes mellitus, autoimmune diseases, and chronic renal diseases, while in the follow-up study patients with hypertension and dyslipidaemia were also included. As a result, in the original study there were 300/1500 participants with comorbidities, while in the follow-up study there are 472/1098. Thus, there are at least 172 patients with hypertension/dyslipidaemia as the only comorbidity. As these conditions per se are not clearly risk factors for HZ/PHN, pooling them with high-risk condition may dilute the effect of comorbidities on vaccine efficacy. I suggest that the authors provide detailed data regarding the number of patients with each underlying disease in Table 1.

·        Furthermore, I suggest that each comorbidity should be tested for association with VE (essentially as a risk factor for HZ/PHN) both in univariate and multivariable analysis. Thus, one might be able to dissociate the effect of vaccination from the presence of various risk factors.

·        Another suggested approach is a propensity score matching analysis where the outcome would be HZ/PHN and vaccination would be among the independent variables.

·        The authors have not analysed the occurrence of HZ/PHN (and therefore VE) as a time to event outcome (i.e., using survival curves). They have only compared the overall incidence (number of events) in the vaccinated and unvaccinated participants. This is important as the efficacy of the live zoster vaccine wanes after a few years.

DISCUSSION

The discussion is well written and of appropriate length.

·        In the limitations of the study the authors should elaborate on the issue patient selection.

·        If the authors are unable to analyse VE as time to event outcome, they should include it in the limitations of the study.

REFERENCES

References are appropriate.

There is a problem as in the reference list numbers 6 and 18 are blank. The authors should see to this.

Author Response

METHODS

In the methods section there are some issues to be addressed:

  • The original safety study had included 1500 participants. Of these, the authors have included in the present study 1096. It is not clear whether there was a selection process, or it was a convenience sample, e.g., the authors included all patients which were able to contact. The authors need to clarify this as the method  (if there was one) of selection might have created a biased sample. E.g., patients who had actually suffered HZ/PHN might be more interested in taking part on the follow-up study. Patients who have died early and were not included in the present study might have a smaller probability of presenting HZ or PHN since the follow-up time was shorter.

We really appreciate your important suggestions. According to your recommendation, we have added the sentences “Only participants who were able to contact with us took part in the present study. “(Lines 71-72) in the method and “First, we included only participants of the previous study, who were able to respond to us. This sampling might have created selection bias because patients who had actually suffered HZ/PHN might be more interested in taking part in the follow up study. On the other hand, the number of those who have died early and were not included in the present study was only seven. Therefore, the selection bias derived from early death might be small” (Lines 239-244) in the limitation.

  • The selection of the control group is controversial, as the authors have used relatives of the participants. If possible, the control group should be matched to the case group, e.g., by age.

    Thank you for your comment. We have added the sentence “Second, we have used relatives of the previous participants as control group. Most relatives were the participants’ spouses. Therefore, the control group might not be matched to the case group by sex. However, their age and lifestyles were similar.” (Lines 244-247).

RESULTS

In the results section the following issues should be addressed:

  • A significant issue inherent with the case-control design, is whether the vaccinated and unvaccinated cohorts are similar in terms of known risk factors for HZ/PHN.
  • The authors have provided adjusted estimates of vaccine efficacy for age, sex, and comorbidities, but they have pooled all comorbidities together. However, not all underlying diseases confer the same risk of HS/PHN and patients with a single low risk comorbidity are considered similar to patients with multiple high-risk comorbidities (e.g., diabetes, CKD and malignancy). It is noted that, in the original study, the authors have included participants with high-risk conditions for HZ, such as malignancy, diabetes mellitus, autoimmune diseases, and chronic renal diseases, while in the follow-up study patients with hypertension and dyslipidaemia were also included. As a result, in the original study there were 300/1500 participants with comorbidities, while in the follow-up study there are 472/1098. Thus, there are at least 172 patients with hypertension/dyslipidaemia as the only comorbidity. As these conditions per se are not clearly risk factors for HZ/PHN, pooling them with high-risk condition may dilute the effect of comorbidities on vaccine efficacy. I suggest that the authors provide detailed data regarding the number of patients with each underlying disease in Table 1.

    Thank you for your important suggestions. We have added the detailed data regarding the number of patients with each underlying disease in Table 1.

  •   Furthermore, I suggest that each comorbidity should be tested for association with VE (essentially as a risk factor for HZ/PHN) both in univariate and multivariable analysis. Thus, one might be able to dissociate the effect of vaccination from the presence of various risk factors.
  • Another suggested approach is a propensity score matching analysis where the outcome would be HZ/PHN and vaccination would be among the independent variables.

    Thank you for your critical suggestion. We have tested each comorbidity for association with occurrence of HZ/PHN by univariate and multivariable analysis. We could not find any significant risk factor that had dissociated VE. Therefore, we have added the sentences “Comorbidities include various diseases such as hypertension, dyslipidemia, diabetes mellitus, chronic renal diseases, autoimmune diseases, respiratory diseases, malignant solid tumor and other diseases. We also conducted analysis with separately controlled for each comorbidity instead of all comorbidities.” (Lines 133-137) and “Even after separately adjusted for each comorbidity instead of all comorbidities, the VEs were not so changed.” (Line 182-183).

  • The authors have not analyzed the occurrence of HZ/PHN (and therefore VE) as a time to event outcome (i.e., using survival curves). They have only compared the overall incidence (number of events) in the vaccinated and unvaccinated participants. This is important as the efficacy of the live zoster vaccine wanes after a few years.

    Thank you for your comment. We have added the sentence “Fifth, we have not analyzed the occurrence of HZ/PHN as a time to event outcome because the hazard ratio of both groups was not constant.”(Lines 256-257) in the limitation. In addition, we have described the median follow-up period in the results section (Lines 155-156).

DISCUSSION

The discussion is well written and of appropriate length.

  • In the limitations of the study the authors should elaborate on the issue patient selection.

Thank you for your comment. We have elaborated the issue patient selection (Lines 239-247) in the limitation.

  • If the authors are unable to analyse VE as time to event outcome, they should include it in the limitations of the study.

Thank you for your comment. We have added the sentence “Fifth, we have not analyzed the occurrence of HZ/PHN as a time to event outcome because the hazard ratio of both groups was not constant.”(Lines 256-257) in the limitation.

REFERENCES

References are appropriate.

There is a problem as in the reference list numbers 6 and 18 are blank. The authors should see to this.

Thank you for your comment. We have corrected the list.

Reviewer 2 Report

overall its important study for to demonstrate the VE for HZ vaccine. 

Shingrix is another vaccine available for HZ. Pls include this in your introdcution and discussion 

1. Please provide detailed sample size calculations to measure effectiveness of the vaccine. Please also describe how (Sample size) was calculated for high risk patients. 

2. Please provide details of how questionnaire/ instrument was validated? Any content validation, pre-testing conducted to decide the appropriateness? 

results:

1. Authors state that adjusted VE 41% (-6 to 67%) was marginal significance (p = 0.08). If your pre-specified p value is < 0.05 and your CI is above and below 0 then you should state your conclusions that adjusted VE is not statistically significant. Please revise the results and conclusions. 

Authors need to revise the discussion to make it clear that crude VE was significant but adjusted VE was not significant.

Limitations:

Authors need to elaborate on the limitation section and include issues related to small sample size, development of questionnaire, follow up period, recall biases, etc 

Author Response

overall its important study for to demonstrate the VE for HZ vaccine. 

Shingrix is another vaccine available for HZ. Pls include this in your introduction and discussion 

  1. Please provide detailed sample size calculations to measure effectiveness of the vaccine. Please also describe how (Sample size) was calculated for high risk patients. 

Thank you for your critical suggestion. If we assumed that VE was 50%, 2% of vaccinated group developed HZ, and 65% of all subjects was vaccinated people, the ideal sample size was more than 2500 subjects. However, we could not recruit enough subjects because this study was an extension of the previous study. We have added the sentence “Sixth, the present study was an extension of a previous study. The sample size might be small to evaluate vaccine effectiveness. Especially, we could not estimate VE against PHN precisely due to few outcomes” (Lines 257-259).

  1. Please provide details of how questionnaire/ instrument was validated? Any content validation, pre-testing conducted to decide the appropriateness? 

Thank you for your comment. If there was a history of HZ diagnosis in the self-administered questionnaire, we asked the medical institution where HZ was diagnosed, to fill in another questionnaire about the details of the disease for validation, as shown in Lines 111-116. The consistency between the self-administered questionnaire and medical records was 81% in the vaccinated group and 90% in the unvaccinated group, as shown in the results section (Lines 168-170).

 results:

  1. Authors state that adjusted VE 41% (-6 to 67%) was marginal significance (p = 0.08). If your pre-specified p value is < 0.05 and your CI is above and below 0 then you should state your conclusions that adjusted VE is not statistically significant. Please revise the results and conclusions. 

Thank you for your comment. We have added the definition of marginal significance in the Methods section (Lines 143-144), in order not to miss the important association because of the small sample size. We have added the sentence “However, the adjusted VE was 41% and could not reach statistically significant.” (Lines 204-205)

Authors need to revise the discussion to make it clear that crude VE was significant but adjusted VE was not significant.

 We appreciated your suggestions. We have referred to the difference of crude and adjusted VE as “the crude overall effectiveness of freeze-dried live attenuated varicella vaccine against HZ was 45% with statistically significance. However, the adjusted VE was 41% and could not reach statistically significant. The VE was similar regardless of comorbidities, sex, and age groups and also statically insignificant. The reason might be small sample size.” (Lines 203-207)

Limitations:

Authors need to elaborate on the limitation section and include issues related to small sample size, development of questionnaire, follow up period, recall biases, et

Thank you for your comment. We have referred to the limitation of the small sample and have added the sentence “Sixth, the present study was an extension of a previous study. The sample size might be small to evaluate vaccine effectiveness.” (Lines 257-259).

We could include the follow up period in the analysis. We have added the sentence “Fifth, we have not analyzed the occurrence of HZ/PHN as a time to event outcome because the hazard ratio of both groups was not constant.” (Lines 256-257).

Regarding the recall biases, we have added the sentence “Seventh, vaccinated people who might be interested in their own health tended to recall their comorbidities and medication in detail. Therefore, there might be recall biases.” (Lines 259-261)

Round 2

Reviewer 1 Report

The authors have responded appropriately to all my queries, except for one.

I need a clarification regarding the survival (time to event) analysis. The authors have stated that they "have not analyzed the occurrence of HZ/PHN as a time to event outcome because the hazard ratio of both groups was not constant".

I suggest that, provided that the authors have the data, to perform the survival analysis. Even though hazards themselves may vary continuously, a simplifying assumption is often made that the hazard ratio is constant across the follow-up period. This is known as the "proportional hazards assumption" (see Cochrane Handbook for Systematic Reviews of Interventions, # 6.8 Time-to-event data https://training.cochrane.org/handbook/current/chapter-06#section-6-8 ).

Author Response

The authors have responded appropriately to all my queries, except for one.

I need a clarification regarding the survival (time to event) analysis. The authors have stated that they "have not analyzed the occurrence of HZ/PHN as a time to event outcome because the hazard ratio of both groups was not constant".

I suggest that, provided that the authors have the data, to perform the survival analysis. Even though hazards themselves may vary continuously, a simplifying assumption is often made that the hazard ratio is constant across the follow-up period. This is known as the "proportional hazards assumption" (see Cochrane Handbook for Systematic Reviews of Interventions,#6.8Time-to-eventdata https://training.cochrane.org/handbook/current/chapter-06#section-6-8 ).

Thank you for your important comment.

At first, we tried to use Cox proportional hazard model to estimate VE but the proportionality was not satisfied. Therefore, we decided to use logistic regression model. The adjusted VE was similar in both models. We have described the history of analysis.

We have added the sentences “To estimate VE against HZ, Cox proportional hazard model was firstly employed. The follow up period was between April 2017 and the onset of HZ or the enrollment of the present study, whichever came first. In this model, we tested the proportional hazards assumption. If p-value suggested that proportionality was not satisfied, we created Kaplan Meier curve to examine the applicability of Cox proportional hazard model. As a result, Cox proportional hazard model was not applicable. Therefore, we decided to use logistic regression model.” (Lines 172-178) in the materials and methods.

We have added the sentences “According to Cox proportional hazard model, the adjusted VE against HZ was 38% (-10 to 66%) but the proportionality was not satisfied (p=0.03).” (Lines 233-234) in the result.

We have added the sentences “Fifth, to estimate VE, we mainly used logistic model instead of Cox proportional model, because the proportional hazard assumption was violated. However, the adjusted VE against HZ was similar in both models.” (Lines 319-321)

Reviewer 2 Report

i am satisfied with revised version 

Author Response

Thank you for your review.